# Variable patterns of mutation density among Na$_V$1.1, Na$_V$1.2 and Na$_V$1.6 point to channel-specific functional differences associated with childhood epilepsy

Alejandra C. Encinas[1☯], Joseph C. Watkins[2☯], Iris Arenas Longoria[2¤], J. P. Johnson, Jr[3], Michael F. Hammer[ID][4]*

**1** Graduate Program in Genetics, University of Arizona, Tucson, Arizona, United States of America, **2** Department of Mathematics, University of Arizona, Tucson, Arizona, United States of America, **3** Xenon Pharmaceuticals, Burnaby, BC, Canada, **4** Department of Neurology, University of Arizona, Tucson, Arizona, United States of America

☯ These authors contributed equally to this work.
¤ Current address: Department of Mathematics, Queen's University, Kingston, Canada
* mfh@email.arizona.edu

## Abstract

Variants implicated in childhood epilepsy have been identified in all four voltage-gated sodium channels that initiate action potentials in the central nervous system. Previous research has focused on the functional effects of particular variants within the most studied of these channels (Na$_V$1.1, Na$_V$1.2 and Na$_V$1.6); however, there have been few comparative studies across channels to infer the impact of mutations in patients with epilepsy. Here we compare patterns of variation in patient and public databases to test the hypothesis that regions of known functional significance within voltage-gated sodium (Na$_V$) channels have an increased burden of deleterious variants. We assessed mutational burden in different regions of the Na$_V$ channels by (1) performing Fisher exact tests on odds ratios to infer excess variants in domains, segments, and loops of each channel in patient databases *versus* public "control" databases, and (2) comparing the cumulative distribution of variant sites along DNA sequences of each gene in patient and public databases (i.e., independent of protein structure). Patient variant density was concordant among channels in regions known to play a role in channel function, with statistically significant higher patient variant density in S4-S6 and DIII-DIV and an excess of public variants in SI-S3, DI-DII, DII-DIII. On the other hand, channel-specific patterns of patient burden were found in the Na$_V$1.6 inactivation gate and Na$_V$1.1 S5-S6 linkers, while Na$_V$1.2 and Na$_V$1.6 S4-S5 linkers and S5 segments shared patient variant patterns that contrasted with those in Na$_V$1.1. These different patterns may reflect different roles played by the Na$_V$1.6 inactivation gate in action potential propagation, and by Na$_V$1.1 S5-S6 linkers in loss of function and haploinsufficiency. Interestingly, Na$_V$1.2 and Na$_V$1.6 both lack amino acid substitutions over significantly long stretches in both the patient and public databases suggesting that new mutations in these regions may cause embryonic lethality or a non-epileptic disease phenotype.

**Data Availability Statement:** All relevant data are within the manuscript and its Supporting Information files.

**Funding:** National Institutes of Health grant GM084905 for a graduate research fellowship to ACE (https://www.nih.gov/) National Science Foundation grant 1740858 to JCW (https://www.nsf.gov/) Xenon Pharmaceutical, Inc provided financial support for a graduate research assistantship to ACE. Commercial funding from Xenon to support graduate research also did not alter our adherence to PLOS ONE policies on sharing data and materials. The funders played no role in the study design, data collection and analysis, decision to publish, or preparation of the manuscript.

**Competing interests:** The authors have declared that no competing interests exist.

## Introduction

Variants in all four brain-expressed voltage-gated sodium channels, $Na_V1.1$, $Na_V1.2$, $Na_V1.3$, and $Na_V1.6$, have been associated with epilepsy [1–3]. While these genes are widely expressed in the cerebral cortex, deep brain nuclei, hippocampus, and cerebellum, their expression patterns differ [4]. $Na_V1.1$ is predominantly localized to the proximal dendrites and soma of excitatory neurons, and the axon-initiating segment (AIS) of fast-spiking parvalbumin-positive inhibitory neurons. This channel is believed to play a major role in controlling network excitability through the activation of inhibitory circuits [1]. Also localized in dendrites and soma, $Na_V1.2$ is expressed in the proximal AIS and in axons of unmyelinated neurons [1]. This channel is predominantly expressed in the neocortex and hippocampus in excitatory neurons, yet has also been reported in somatostatin-positive inhibitory interneurons [3]. Shortly after birth, $Na_V1.6$ is expressed at nodes of Ranvier in multiple neuronal classes and glia across the cortex, hippocampus, brain stem and cerebellum [1, 4]. Highly expressed in the central nervous system (CNS), $Na_V1.6$ is concentrated at the AIS in both excitatory and inhibitory neurons, and at nodes of Ranvier in myelinated neurons, where it mediates the initiation and propagation of action potentials [1]. Mutations in $Na_V1.1$ have been associated with epilepsy exhibiting a wide spectrum of severity, including Dravet syndrome and genetic epilepsy with febrile seizures plus (GEFS+) [1, 3]. Mutations in $Na_V1.2$ have also been associated with GEFS+ and benign familial neonatal-infantile seizures (BFNIS), as well as autism and a more severe form of epileptic encephalopathy [1, 3]. Also having a broad range of severity [5], variants in $Na_V1.6$ were initially found to be associated with an epileptic encephalopathy (EIEE13) characterized by intellectual disability and developmental delay [6].

Knowledge of the roles that $Na_V1.1$, $Na_V1.2$, and $Na_V1.6$ channels play in epilepsy has increased greatly in the past decade, yet the ability to predict the clinical outcome of a variant in any of these channels remains an unmet and important challenge. To aid in the interpretation of the pathological significance of $Na_V$ variants, we sought to investigate associations across the protein between functionality and the distribution of variants using both public and patient databases. Previous literature has focused on the variant distribution within a single channel to evaluate channel specific properties, we however, are among the first to analyze the variant distribution of three $Na_V$ channels to find both common and unique patterns among these three sodium channels.

## Materials and methods

### Variant database

We made use of a database of $Na_V1.1$ missense variants from 758 patients with SCN1A-related epilepsy [7] including 661 with Dravet Syndrome [8]. $Na_V1.2$ missense variants from Wolff et. al. [9] were included in our analysis, as were additional variants found in a PubMed literature search between June 2016 and October 2018 utilizing the term 'scn2a'. For $Na_V1.6$ mutations, we included 54 variants from the SCN8A registry [10], as well as 70 additional published variants. For sequence comparisons, $Na_V1.1$, 1.2, and 1.6 sequences were aligned with Uniprot. The Institutional Review Board at the University of Arizona approved the SCN8A registry. The human subjects committee approved an online informed consenting process. Informed consent for minors was obtained from parents. This study also used data generated by DECIPHER (http://decipher.sanger.ac.uk). To compare distributions of variants in the above-mentioned databases with those from individuals not known to be affected with pediatric disease, we utilized the Genome Aggregation database (gnomAD), which includes 123,136 exomes and 15,496 genomes from a total of 138,632 individuals [11]. The total number of patient variants

and gnomAD variants (here termed 'public') is listed in S1 Table and a list of all patient variants used in this study can be found in S2 Table.

### Definition of low and high functionality regions in channels

A summary of the functionality of specific $Na_V$ regions [12] is presented in S3 Table. Ishii et al. [13] noted a relatively sharp boundary between the first and second halves of both the N- and C- termini of *SCN1A* Dravet syndrome missense variants, which provided motivation to separately investigate each region of the $Na_V$ channel termini in this study. The S3-S4 linker has been shown to contain binding sites for channel modulators [14] and therefore was not placed in either functionality group. The remaining regions were placed into the high functionality category if disruption of their function could cause a foreseeable impact on the rate at which sodium ($Na^+$) ions move into the cell.

### Data analysis

Fisher tests were calculated for each $Na_V$ channel region comparing the proportion of patient *versus* public variants included in the gnomAD database in that particular region. To avoid issues with multiple comparisons we focused our discussion of significant results for those cases where the p-value was below 0.0001. The Fisher test provided p-values for odds ratios, which were calculated for each $Na_V$ channel region. The Anderson-Darling test was used to provide a measure of agreement among the variant cumulative distribution functions. The empirical cumulative distribution of variant sites along the DNA sequence of each sodium channel provides insights into common properties of a segment that may not necessarily adhere to the protein structure of segments, linkers, loops and termini. Moreover, all of the Fisher tests and the Anderson-Darling tests can be computed from the cumulative distributions. Under an assumption of uniformly distributed variants, if a contiguous sequence of cDNA had a lower than 1% probability of containing no variants, then the section was designated "flat". Under the same assumptions we constructed a "lethality test" to test against a null hypothesis of non-lethality. Those with an excess of variants were denoted as "steep". This results in four possible labels for a given contiguous sequence of amino acids for comparisons between patient and public databases; lethal = flat/flat, pathogenic = steep/flat, complex = steep/steep, and benign = flat/steep.

For those sections that had no variants in either the public or patient databases (flat/flat), the absence of variants leads to the presumption that variants in this section are lethal or are associated with a phenotype that precludes inclusion in either the patient or public databases. If we encounter a section that has only patient variants (steep/flat), we call the segment pathogenic. Sections that have only mutations in public individuals (flat/steep) are likely tolerant to variants. Finally, if a section has variants from both groups (steep/steep), then a more complex explanation is necessary to describe the pattern of variant distribution (e.g., mingling of pathogenic and benign variants).

## Results

### $Na_V$ divergence

All three $Na_V$ channels are similar in their structure and overall sequence with 74% amino acid similarity among the channels. The percent of amino acid sequence difference among all three channels with the majority of regions having less than 30% divergence and only a handful of regions having higher than 30% divergence in the amino acid sequence (the termini, S5-S6$_{DI + DIII}$, DI-DII, S1-S2$_{DII+DV}$, DII-DIII, S2-S3$_{DII}$, and S2$_{DIV}$) for all three channels (data not

shown). Indels were restricted to seven regions in the channel (C and N termini, S5-S6$_{DI, DIII, DIV}$, DI-DII, and DII-DIII).

## Variant density in low and high functionality regions

We counted the number of variants in each segment, loop, and linker in all three sodium channels relative to the total number of variants in the patient and public databases (data not shown). The majority of variants in the public databases were located in the loops (DI-DII and DII-DIII), followed by the C-terminus and the N-terminus. We rejected the null hypothesis of a uniform distribution for each of the Na$_V$ channels in both the patient and public databases (Anderson-Darling: Na$_V$1.1: patient p = 1.12 x 10$^{-6}$, public p = 9.69 x 10$^{-7}$; Na$_V$1.2: patient p = 3.03 x 10$^{-6}$, public p = 1.23 x 10$^{-6}$; and Na$_V$1.6: patient p = 4.84 x 10$^{-6}$, public p = 1.54 x 10$^{-6}$). Fig 1 depicts the cumulative distribution of both patient and public data for Na$_V$1.1 as an example (see S1 and S2 Figs for the cumulative distributions of Na$_V$1.2 and Na$_V$1.6 variants). Additionally, as a measure of similarity of the distributions, we conducted pairwise tests between patient databases (Anderson-Darling: Na$_V$1.1 *versus* Na$_V$1.2 p = 0.40; Na$_V$1.2 *versus* Na$_V$1.6 p = 0.21; Na$_V$1.1 *versus* Na$_V$1.6 p = 0.30 and public databases (Anderson-Darling: Na$_V$1.1 *versus* Na$_V$1.6 p = 2.02 x 10$^{-2}$; Na$_V$1.1 *versus* Na$_V$1.2 p = 0.11; Na$_V$1.2 *versus* 1.6 p = 0.33). Thus, Na$_V$1.2 and Na$_V$1.6 show greater overall similarity in the distribution of variants in the public databases.

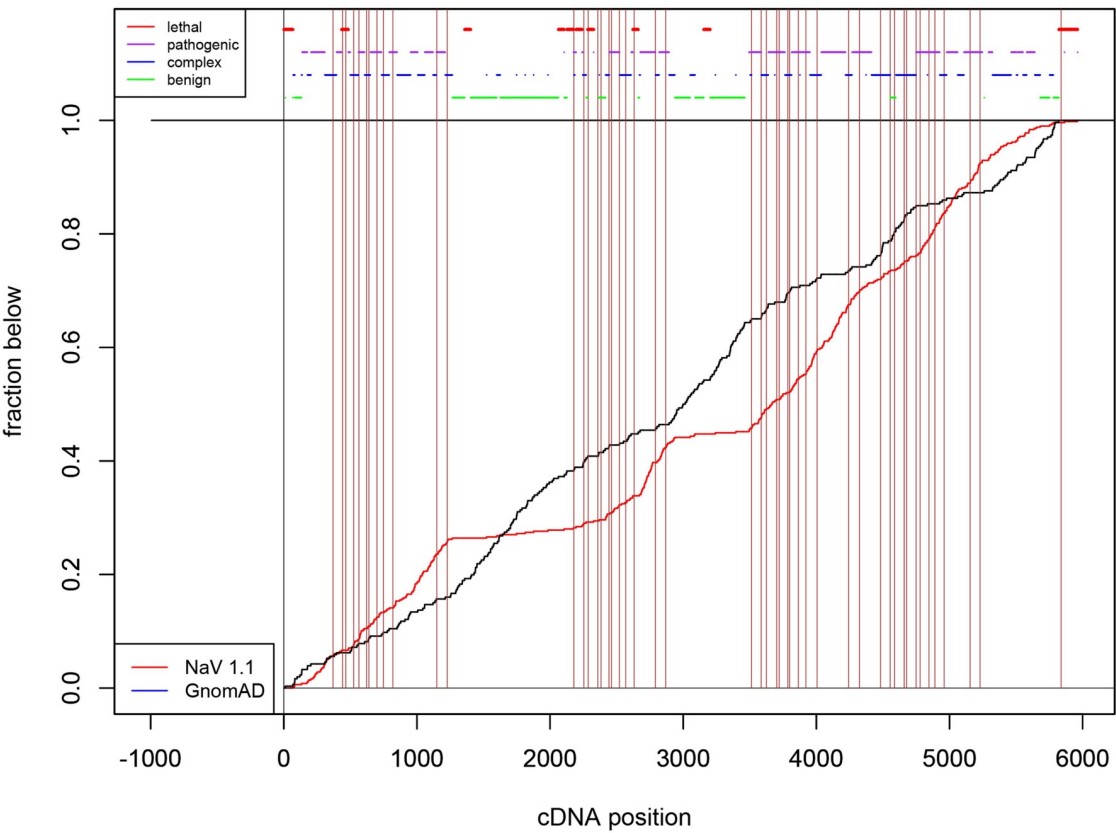

**Fig 1. Na$_V$1.1, 1.2, and 1.6 sequences were aligned with Uniprot and the percent of amino acid residues that differ between all three channels (black bar-right).** The overall percent of patient (red) and GnomAD (black) mutations combined for Na$_V$1.1, Na$_V$1.2, and Na$_V$1.6 are displayed on the left side of the figure.

In testing for the relative excess of patient or public variants in different channel regions when simultaneously comparing all three channels, we find that the variant distributions between low and high functionality regions are strongly significantly different (Mantel-Haenszel test p = 2.2 x $10^{-16}$). For pairwise comparisons, we performed a series of Fisher tests on odds ratios (OR). In considering the entire gene, this ratio was higher (OR = 19.07; Fisher test p = 2.2 x $10^{-16}$) for $Na_V1.1$ patient variants in high functionality regions. The high functionality pattern also exists in $Na_V1.2$ (OR = 10.56; Fisher test p = 2.90 x $10^{-7}$) and $Na_V1.6$ (OR = 26.01; Fisher test p = 1.37 x $10^{-6}$). Correspondingly, it follows that public variants had a higher OR in low functionality regions.

We then conducted a similar analysis on low functionality loops, segments, and linkers. We found a higher OR for public variants within the DI-DII loop of $Na_V1.1$, $Na_V1.2$, and $Na_V1.6$ (OR = 15.14, 6.01, 6.69 respectively; Fisher tests p = 2.2 x $10^{-16}$ in all cases). Public variants also had a higher OR in the DII-DIII loop for $Na_V1.1$, $Na_V1.2$, and $Na_V1.6$ (OR = 9.46, 4.67, 7.91 respectively; Fisher tests, p = 2.2 x $10^{-16}$, 6.03 x $10^{-8}$, and 8.35 x $10^{-7}$ respectively). Interestingly, patient variants had 3.53 higher OR within S3 segments for $Na_V1.6$ (Fisher test, p = 7.4 x $10^{-3}$). The remaining segments and linkers in the low functionality group had OR that were not found to be statistically significant.

The high functionality regions showed higher concordance among the channels, with patient variants having a higher OR in S4, S5, and S6 segments and the S4-S5 linker. $Na_V1.1$ variants in the S5-S6 linker had an OR that was 11.82-fold higher for patient variants (Fisher test, p = 2.2 x $10^{-16}$) and only $Na_V1.6$ had a higher OR of patient variants within the inactivation gate (OR = 18.05; Fisher test, p = 8.19 x $10^{-6}$) (Table 1 and S4 Table).

## Variant density by domain

Heat maps displaying log odds ratios for each channel are shown in Fig 2. Within each domain no transmembrane segments had statistically significantly higher OR for public variants, while patient variants had a higher OR within segments and linkers situated beyond segment 3 (Fig 2). For all channels, the hotspots for higher patient variant OR included S4, S4-S5 linker, and S5 (Fig 2). $Na_V1.1$ was also found to have significantly higher patient variant OR in $S3_{DI}$ (OR = 51.85 Fisher test p = 7.42 x $10^{-10}$) and $S1_{DIII}$ (OR = 10.54; Fisher test p = 1.26 x $10^{-2}$) (Fig 2A), while patient variant OR in $Na_V1.2$ and 1.6 were found to be significantly higher for $S3_{DIV}$ (OR = 5.92, 4.59, respectively; Fisher test p = 7.91 x $10^{-3}$, 1.08 x $10^{-2}$, respectively) (Fig 2B/2C). The S4-S5 linker had statistically significantly higher patient variant OR for $Na_V1.1$ in DIV (OR = Infinite; Fisher test p = 8.036 x $10^{-13}$) while patient variant OR for $Na_V1.2$ were higher in DII, DIII and DIV (OR = Infinite, 7.69, Infinite, respectively; Fisher test p = 1.53 x $10^{-5}$, 1.11 x $10^{-3}$, 1.92 x $10^{-3}$, respectively), and for $Na_V1.6$ were higher for all four domains

**Table 1. Statistically significant OR values for patient and public variant density for termini and segments within $Na_V1.1$, $Na_V1.2$, and $Na_V1.6$.**

| Region | | N-term | S1-S3 | S3-S4 | S4 | S4-S5 | S5 | S5-S6 | S6 | DI-DII | DII-DIII | DIII-DIV | C-term |
|---|---|---|---|---|---|---|---|---|---|---|---|---|---|
| **Functionality**[a] | $Na_V$ | na | Low | na | High | High | High | High | High | Low | Low | High | na |
| **Higher Patient OR** | **1.1** | | | - | X | X | X | X | X | | | | |
| | **1.2** | | | | X | X | X | | X | | | | |
| | **1.6** | | | | X | X | X | | X | | | X | |
| **Higher Public OR** | **1.1** | | | - | | | | | | X | X | | X |
| | **1.2** | X | | - | | | | | | X | X | | X |
| | **1.6** | X | X (S3 only) | - | | | | | | X | X | | X |

[a] see S3 Table. X indicates Fisher test p-value <0.0001.

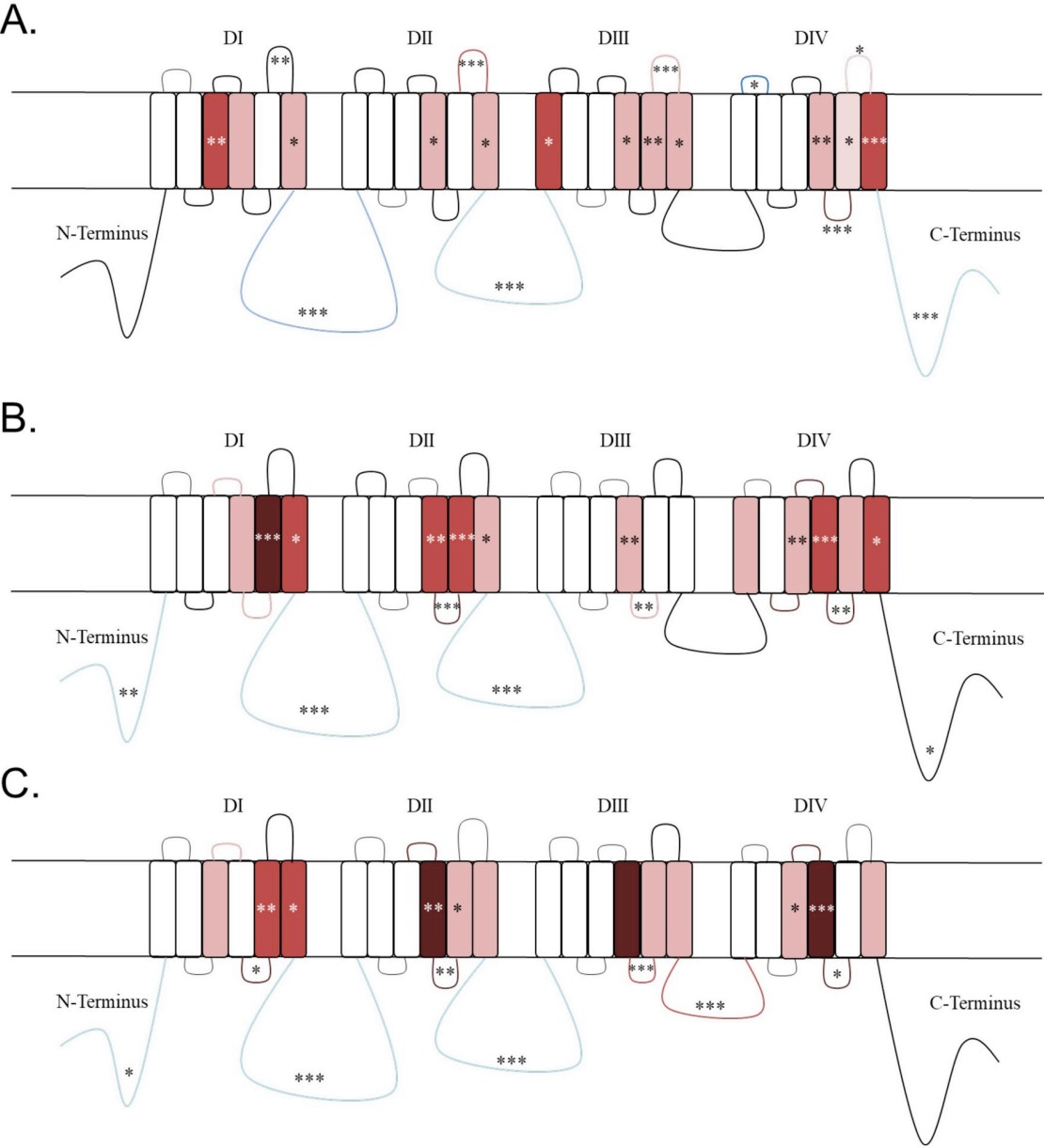

**Fig 2. Heat map for A) Na$_V$1.1, B) Na$_V$1.2, and C) Na$_V$1.6.** Significant p-values on the heat maps were indicated as follows: ***0.001, ** = 0.001–0.01, * = 0.01–0.05. Regions are color-coordinated in increasing darkness to indicate increased logarithmic odd ratio values. Red indicates pathogenic regions, while benign regions are displayed in blue.

(DI, DII, DIII, DIV: OR = Infinite, Infinite, 23.12, Infinite, respectively; Fisher test p = 1.38 x $10^{-2}$, 1.38 x $10^{-2}$, 2.66 x $10^{-4}$, 1.38 x $10^{-2}$, respectively).

### Distributional patterns of variants in loops and termini

We used cumulative distribution analyses to examine patterns of variant distribution within the intracellular loops and the N- and C-termini for all three channels (Fig 1, S1 and S2 Figs). The biphasic pattern previously reported for the N-terminus of Na$_V$1.1 [13] was also apparent for Dravet patients in the current analysis; i.e., the patient/public slopes exhibited a benign

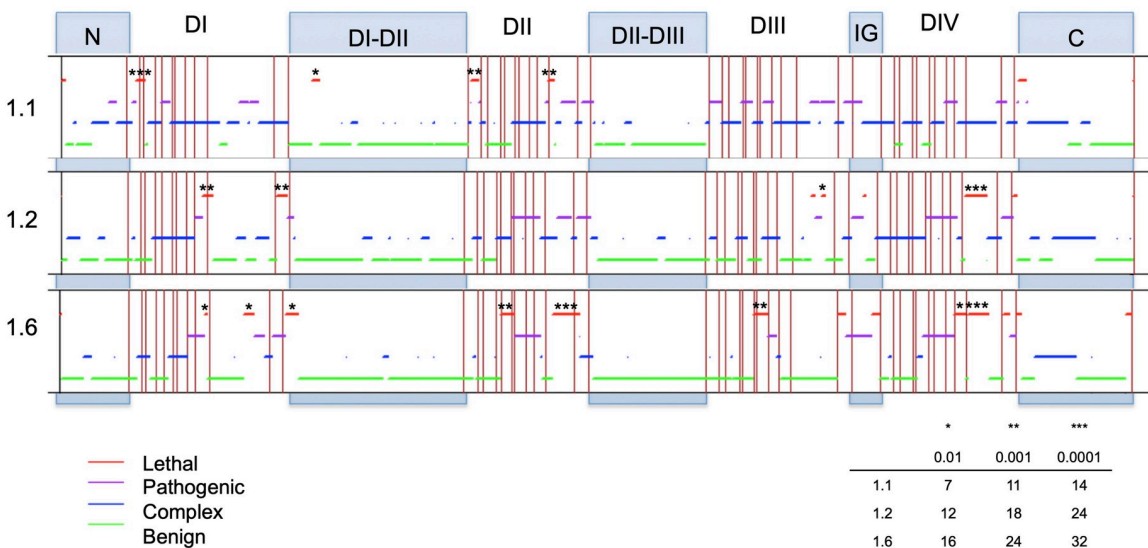

**Fig 3. Cumulative distribution functions for Na$_V$1.1, 1.2, and 1.6 missense mutations.** Comparison of public mutations (black line) and patient missense mutations (red line). The vertical lines represent the segment and domain boundaries for each channel. The table shows the thresholds for lack of variants to obtain the given p-values for the lethality test.

(i.e., flat/steep) pattern in the 5'end and a complex (steep/steep) pattern at the 3'end of the N-terminus. In contrast, there was a complex (i.e., steep/steep) pattern in the 5' end of the C-terminus and a benign pattern in the 3' end of the C-terminus (Fig 3). For Na$_V$1.2, there was not a simple biphasic pattern in either the N- or C- terminus; however, in general the N-terminus was mostly benign while in the C-terminus there was an 'island' of sequence with a complex pattern in the midst of sequence characterized by the benign pattern for most of the remaining C-terminus. The Na$_V$1.6 N- and C-termini had a very similar pattern as Na$_V$1.2.

The first two intracellular loops for all three channels were dominated by a benign pattern; albeit with varying numbers of short stretches of complex patterns dotting the loop sequences (Fig 3). Unlike the patterns in Na$_V$1.1 or Na$_V$1.2, the 5' two-thirds of the Na$_V$1.6 inactivation gate was characterized by a pathogenic (i.e., steep/flat) pattern that began in S6$_{DIII}$. This region was flanked by sequence exhibiting the lethal (i.e, flat/flat) pattern in S6$_{DIII}$ and in the last third of the inactivation gate (Fig 3).

Using the suggestion in Ishii et al. [13], we divide the Na$_V$1.1 N- and C-termini into sections according to the boundaries of high and low variant density observed in the cumulative distribution analysis and re-confirm several strongly significant results; the first 57aa contained a larger number of public *versus* patient variants (2 patient and 30 public) (OR = 0.08, Fisher test p = 4.56 x 10$^{-6}$), while the remaining 66aa had a larger number of patient *versus* public variants (42 patient and 16 public) (OR = 9.52, Fisher test p = 4.86 x 10$^{-16}$. The C-terminus displayed the opposite pattern with the initial 100 aa containing a larger number of patient variants (23 patient and 36 public) (OR = 2.78, Fisher test p = 7.77 x 10$^{-6}$) and the remaining 123 aa with a higher number of public variants (3 patient and 68 public) (OR = 0.05, Fisher test p = 1.83 x 10$^{-14}$).

Interestingly, this pattern is less strong at the extreme parts of the termini upon the addition of the 97 mild Na$_V$1.1 variants. For the first 57aa (2 patient and 29 public) the OR increases to 0.19 (p = 0.012). The distinction in variant density remains very strong for the remaining 66aa variants (46 patient and 16 public) (OR = 9.07, Fisher test p = 3.13 x 10$^{-16}$. Similarly, for the C terminus, we have for the initial 100 aa (29 patient and 36 public variants) we have OR = 2.84

(p = 6.47 x $10^{-5}$). For the remaining 123 aa, six mild variants give a total of 9 patient variants and 68 public variants (OR 2.24, p = 1.89 x $10^{-3}$). The comparative enrichment of mild variants in the termini can be seen in the comparison of cumulative distribution function (S3 Fig).

A similar but much weaker pattern emerged when we divided the $Na_V1.2$ and $Na_V1.6$ N- and C- termini sequences. The most significant result in this region was the 3'end of the $Na_V1.6$ C-terminus, which had a 102aa stretch of mainly public variants (1 patient and 32 public) (OR = 0.091; Fisher test p = 1.37 x $10^{-3}$).

For the inactivation gate, the only strongly significant finding across all three channels was an excess of patient variants in the first 26 aa in $Na_V1.6$ (8 patient and 0 public) (OR = Infinite; Fisher test p = 1.31 x $10^{-5}$). Interestingly, we did not observe a similar signal in $Na_V1.2$.

## Distributional patterns of variants within domains

Regions of lethality were present in all three channels (Fig 3). These represent regions that would not become apparent when conducting Fisher tests because they contain no patient or public variants. In $Na_V1.1$, statistically significant lethal regions were present in $S1_{DI}$ to $S2_{DI}$ (aa length = 16; lethality test p = 5.54 x $10^{-4}$), $S1_{DII}$ (aa length = 14; lethality test p = 1.42 x $10^{-3}$), $S5-S6_{DII}$ (aa length = 13; lethality test p = 2.27 x $10^{-3}$), and the 3' stretch of the C-terminus (aa length = 15; lethality test p = 8.87 x $10^{-4}$) (Fig 3). The longest lethal pattern in $Na_V1.1$ occurred in the $S1_{DI}$ to $S2_{DI}$ region (16 aa) and was 87.5% (14/16) conserved between all three channels. In $Na_V1.2$, the statistically significant lethal regions occurred in $S5_{DI}$ to $S5-S6_{DI}$ (aa length = 19; lethality test p = 6.84 x $10^{-4}$), $S6_{DI}$ (aa length = 18; lethality test p = 1.00 x $10^{-3}$), and the inactivation gate (aa length = 40; lethality test p = 2.17 x $10^{-7}$) (Fig 3). The longest stretch of lethality for $Na_V1.2$ occurred in the $S5-S6_{DIV}$ region which is highly conserved among all three channels (92.5%–37/40). For $Na_V1.6$, statistically significant lethal regions were on average longer and were present in $S2-S3_{DII}$ to $S3_{DII}$ (aa length = 25; lethality test p = 8.24 x $10^{-4}$), $S5-S6_{DII}$ to $S6_{DII}$ (aa length = 50; lethality test p = 6.79 x $10^{-7}$), $S3-S4_{DIII}$ to $S4_{DIII}$ (aa length = 27; lethality test p = 4.67 x $10^{-4}$), and $S5-S6_{DIV}$ (aa length = 37; lethality test p = 2.73 x $10^{-5}$) (Fig 3). The longest stretch of lethality for $Na_V1.6$ occurred from the S5-S6 linker to S6 segment in domain DII. This 50 aa stretch is highly conserved among all three channels (90%–45/50).

Additionally, there were two instances in which the lethal regions between channels overlapped. The first occurred in the $S5-S6_{DIV}$ linker between $Na_V1.2$ and $Na_V1.6$, which were both statistically significant lethal regions (Fig 4A). The second instance occurred in the C-terminus where $Na_V1.2$ and $Na_V1.6$ individually overlapped with $Na_V1.1$ lethality; however, the stretch of lethality was only found to be significant for $Na_V1.1$ (Fig 4B). The lethal region of $Na_V1.1$ overlapped with $Na_V1.6$ on the seventh aa into the C-terminus and spanned 4 aa (Fig 4B). The overlap for $Na_V1.1$ and $Na_V1.2$ occurred at the 13$^{th}$ aa position and spanned 8 aa. The 2 aa

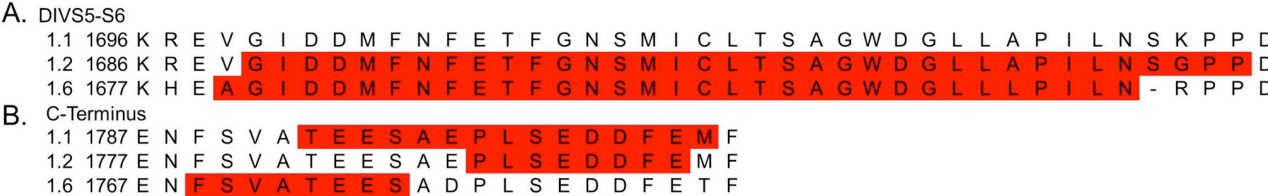

**Fig 4. The amino acid sequences for all three channels are shown.** These sequences are mapped and aligned to show homologous amino acid pairings. Letters marked in gray represent a lethal region as defined by the cumulative distribution plots. A) The first 41 amino acid bases for the $S5-S6_{DIV}$ linker. B) The first 22 amino acid bases for the C-terminus.

between the $Na_V1.2$ and $Na_V1.6$ lethal regions were partially conserved between the three channels (Fig 4B). Overall, lethal areas that did not overlap between channels were still found to be highly conserved, and only 13% (47/352) of amino acids had reported patient variants among the three channels.

## Discussion

Voltage-gated sodium ($Na_V$) channels are responsible for the initiation and propagation of action potentials and are specialized for electrical signaling. Humans have nine distinct $Na_V$ channel genes, each of which encodes of a single pore-forming α-subunit composed of four repeat domains (I–IV), which in turn each contain six transmembrane α-helical segments. The four transmembrane domain structure of all $Na_V$ channels is highly conserved [14]. The six sequentially linked transmembrane segments (S1 through S6) of each domain consist of a voltage-sensing domain (VSD: S1 through S4) and a pore-forming domain (PFD: S5 through S6). The VSD is highly flexible and shifts during membrane depolarization [15]. The outward movement of S4 is coupled to the PFD via the S4-S5 linker, and results in the PFD opening in an iris-like manner to allow sodium ions to flow into the cell [15]. The S5-S6 linker contains the selectivity filter (with the four amino acid abbreviation DEKA), which is highly conserved for each $Na_V$ channel and forms the narrowest portion of the pore [14]. The DIII-DIV loop, also known as the inactivation gate, is often depicted as a hinged lid to close the intracellular portion of the pore.

Four voltage-gated sodium channel (VGSC) isoforms are highly expressed in the CNS and are implicated in epilepsy. We investigated patterns of variation among three of these channels with the aim of gleaning new insights into mechanisms of channel pathology. To accomplish this task, we analyzed cohorts of patient and public variants from three well-studied sodium channel systems ($Na_V1.1$, $Na_V1.2$, and $Na_V1.6$). All three channels had excess patient variant burden for S4 to S6, with 77% (302/391), 71% (104/146), and 69% (57/83) of patient variants located in $Na_V1.1$, $Na_V1.2$, and $Na_V1.6$, respectively (S5–S7 Tables). In particular, S4 and S6 showed similar higher patient variant burden as measured by the OR across all three channels, consistent with known functionality of these regions in sodium channels [16–18]. However, we also discovered $Na_V$ channel specific hot spots in the inactivation gate, the S5-S6 linker, the S4-S5 linker, and S5. In the following sections, we discuss the features that distinguish each channel based on its position within the neuron and the type of neuron in which it functions (excitatory and inhibitory).

### $Na_V1.6$-specific burden in inactivation gate may be related to action potential propagation

Proper fast inactivation is necessary for the repetitive firing of action potentials in neurons [18]. The inactivation gate is responsible for the fast inactivation of VGSCs, and performs rapid inactivation by acting as a hinged lid on the intracellular side of the pore [12]. The first half of the inactivation gate contains the previously identified hydrophobic cluster (isoleucine, phenylalanine, and methionine—IFM) that maintains the closed state of the channel via docking sites on $S6_{DIV}$ and $S4-S5_{DIV}$ [19–21]. Gain-of-function (GoF) mutations in the inactivation gate lead to increased persistent current [22], which is known to facilitate repetitive firing. Therefore, an inability to perform fast inactivation would cause increased persistent current leading to increases in hyperexcitability by lowering the activation threshold for subsequent action potentials [23]. Indeed, $Na_V1.6$ was the only channel to display statistically significant patient variant burden for the entire length of the inactivation gate. A closer examination of

the region in $Na_V1.6$ revealed that the first half of the gate contains an especially higher patient variant burden.

This requirement for proper fast inactivation to occur may not produce an epileptic manifestation for $Na_V1.1$ and $Na_V1.2$ due to their location within the neuron. $Na_V1.1$ is hypothesized to play a role in controlling network excitability through activation of inhibitory circuits [1] and $Na_V1.2$ is thought to control back propagation of action potentials [24]. Therefore, improper fast inactivation in these circumstances would not directly lead to an overall hyperexcitability. Experiments that test the effect of amino acid substitutions in the inactivation gate for both $Na_V1.1$ and $Na_V1.2$ will help to better understand the $Na_V1.6$ sensitivity in this region.

## Excess of patient variants in S5-S6 linkers of $Na_V1.1$ may be associated with loss of function

The ability of the $Na_V$ channel to initiate action potentials is due to selective transport of sodium ions across the membrane, which occurs through a pore-forming module (S5-S6) [25]. Each of the S5-S6 linkers forms a P-loop which consists of an extracellular linker to S5, a descending P-helix, an ascending limb, and an extracellular linker to S6 [26]. The ascending portion of the P-loop contains the selectivity filter, which is made up of a single amino acid from each domain, and forms the narrowest portion of the channel [27, 28]. Residues within these linkers have been shown to contribute to proper permeation of $Na_V$ channels [25]. In our study, only $Na_V1.1$ displayed higher patient variant burden in the S5-S6 linker. We hypothesize that patient variants in this region for all VGSCs lead to impaired sodium selectivity, which in turn may produce a loss of channel function. Sodium channels achieve their balance of selectivity and high $Na^+$ flux by precisely aligning the carbonyl oxygens from the peptide backbone in a four-fold symmetry that replaces the water molecules that would normally hydrate a $Na^+$ ion in solution. Any structural change that disrupts the alignment or spacing of the carbonyl groups will reduce the ability of the channel to strip the shell of water from the $Na^+$ and allow the ion to pass through the channel [27]. Depending on the variant, this could also cause a reduction in current without loss of selectivity. Selectivity mutations can cause diverse impacts on function. Many of them will lead to poor/slow calcium or potassium ion permeation and hence reduce sodium ion flux and appear as a loss of function. If the variant allows robust calcium permeation it may actually appear as a GoF since calcium influx is normally downstream of $Na_V$ activation.

## Region-specific variant deserts in patient and public databases: Evidence of lethality or alternate phenotypes?

Fig 3 illustrates the position of 16 lethal regions (i.e., statistically significant long stretches with no amino acid substitutions in either the patient or public databases): 5 in $Na_V1.1$, 4 in $Na_V1.2$ and 7 in $Na_V1.6$. Four of these stretches are shared within the same segment, linker, loop or terminus across two channels. All but one of these shared lethal stretches was present in the S5-S6 segment (i.e., in DI, DII and DIV); and all three involved $Na_V1.6$ and two of the three involved $Na_V1.2$. While we refer to these regions as 'lethal' (i.e., implying that variants cause embryonic lethality), it is also possible that variants in these regions are associated with a non-epilepsy disease phenotype—hence, excluding them from both the epilepsy and public databases. Scrutiny of other patient databases is needed to test the latter possibility.

Given the prevalence of the longer stretches (i.e., $p<0.001$) in S5-S6 segments of $Na_V1.6$ and $Na_V1.2$, we hypothesize that variants in these parts of the channel cause a total loss of function. We know that total loss of function is not generally lethal in $Na_V1.1$ (e.g., Dravet

Syndrome patients survive). On the other hand, total loss of function variants are quite rare within $Na_V1.2$ and $Na_V1.6$ and these variants are often associated with non-epilepsy phenotypes, such as movement disorders, ataxia, intellectual disability and/or autism and may not lead to epilepsy [29, 30]. Indeed, all three channels demonstrate intolerance to loss of function (LoF) variants as reflected by a paucity of nonsense, frameshift, and splice site variants in the gnomAD database. We computed z-scores for tolerance to LoF variants in all VGSCs (S4 Fig) and found that brain-expressed channels ($Na_V1.1$, $Na_V1.2$, $Na_V1.3$, and $Na_V1.6$) were orders of magnitude more intolerant when compared with those that are primarily expressed in peripheral nervous system and muscle (e.g., $Na_V1.7$, $Na_V1.8$, and $Na_V1.9$).

Another possibility is that variants in the lethal stretches have dominant negative effects, which may well be lethal. For example, Berecki et al. [31] suggested that a variant in $Na_V1.1$ (T226M) that is associated with a more severe phenotype than Dravet syndrome [32] leads to a reduction of $Na_V1.1$ current through both the mutant and the wildtype allele so that the sum current is >0 but <50%. Prenatal lethal mutations could be more extreme versions of this phenotype. If this were the case for GoF variants in $Na_V1.2$ and $Na_V1.6$ the loss of any wildtype expression could cause embryonic lethality.

## Variant burden associated with inhibitory versus excitatory neuron pathways

Patterns of variation in the S4-S5 linkers and S5 segments was concordant with the expression of these channels on excitatory *versus* inhibitory neurons. These regions have been previously implicated in causing both hyperexcitability and loss of channel function in a domain-dependent manner. We hypothesize that these patterns are due to the channel effect on each of these differing neuron types. For example, the S4-S5 linker in both domains III and IV make up the portion of the channel necessary for fast inactivation [33]. Mutations in this region have been shown to effect channel function in the muscle-expressed channel $Na_V1.4$ [33]. In DIII the S4-S5 linker is hypothesized to interact with amino acids in $S6_{DIV}$ that transmit movement of $S4_{DIII}$ to $S6_{DIV}$ and plays a role in fast inactivation [34]. While the $S4-S5_{DIV}$ linker was found to interact with the inactivation gate during fast inactivation and mutations in this region were found to disrupt fast inactivation [35]. Our results are consistent with these findings because $Na_V1.2$ and $Na_V1.6$ showed higher patient variant burden in the S4-S5 linker in DIII and DIV. Interestingly, $Na_V1.1$ only shows higher patient variant burden in $S4-S5_{DIV}$, while $Na_V1.6$ shows increased patient variant burden in the S4-S5 linker in DI and DII. The relative paucity of patient variants in the inactivation machinery may imply that GoF Nav1.1 variants are relatively well tolerated. In fact, very few patients with GoF variants have been identified (with the exception hemiplegic migraine). If true, this would suggest that therapies that induce GoF in $Na_V1.1$ could be well tolerated even if they disrupt inactivation as a means of increasing interneuron $Na_V$ current. This population-based variant study shows the need for further investigation into structure-function relationships in sodium channels especially in DI and DII.

Previously, a point mutation in $S5_{DII}$, within $Na_V1.4$, was found to shift the activation curve in the hyperpolarizing direction. This mutation had minor effects on fast inactivation while greatly impairing slow inactivation and allowing for a more rapid activation of the channel [36]. Whereas, a point mutation in $Na_V1.4$ in the $S5_{DIV}$ segment led to this channel becoming activated and inactivated at more negative potentials, ultimately causing slowed recovery from fast inactivation with no effect on channel deactivation [36]. Additionally, $S5_{DIV}$ has been shown to be part of a hydrophobic cavity that interacts with the inactivation gate [37]. Both $Na_V1.2$ and $Na_V1.6$ had a higher patient variant burden in $S5_{DI}$ and $S5_{DII}$, while $Na_V1.1$ had a higher patient variant burden in $S5_{DIII}$ and $S5_{DIV}$. The results of Bendahhou et al. [36] imply

that S5$_{DII}$ variants in Na$_V$1.2 and Na$_V$1.6 would lead to a hyperactive sodium channel, while such variants in Na$_V$1.1 would cause excess firing on inhibitory interneurons. On the other hand, a higher patient variant burden in Na$_V$1.1 S5$_{DIV}$ should slow recovery from inactivation and lead to a subsequent loss of channel function. Future studies should be conducted to fully determine the channel specificity in higher patient variant burden in S5 among all three channels to better understand the potential gating mechanism that is affecting activation and inactivation within these segments.

## Do variants in the C-terminus and DI-DII intracellular loop reflect channel function?

The C-terminus is made up of a globular domain containing a well-structured EF-hand (helix-loop-helix structure) followed by an unstructured extended helix containing the IQ (isoleucine–glutamine) motif, which is known to interact with calmodulin [38, 39]. In Na$_V$1.5, the proximal half of the C-terminus has been shown to interact and stabilize the inactivation gate and mutations within this region have been identified in multiple cardiac rhythm disturbances [20, 39]. The proximal C-terminus contains the interaction site for fibroblast growth factors as well as the binding sites of β1 to β4 sodium channel subunits. Fibroblast growth factor 14 has been suggested to play a key role in the organization of VGSC alpha subunits in the AIS, while the β subunits are essential for the modulation of current and proper expression of VGSCs on the cell surface [39]. Given the established role of the proximal C-terminus in channel inactivation, it is surprising that only Na$_V$1.1 displayed a statistically significantly higher patient variant burden. This suggests that these variants have more impact on trafficking (i.e., resulting in normal protein level in neurons) than on inactivation. The lack of a high patient variant burden in Na$_V$1.2 and Na$_V$1.6 in this region may highlight a higher tolerance of variation within these channels for this region.

Additionally, the distal C-terminus, which is known to house the IQ-motif, is consistent among the three channels as having an excess of public variants. Similarly, the DI-DII loop was also shown to have an excess of public variants. This loop has previously been shown to contain five phosphorylation sites, which are involved in neuromodulation [16]. The statistically significantly low OR for both the distal C-terminus and the DI-DII intracellular loop may indicate that pathogenic variants are limited to the particular sites of interaction (i.e. IQ-motif in the C-terminus and sites of phosphorylation in DI-DII) and not the regions surrounding them. Therefore, even if a region contains more public variants, this does not exclude the possibility that there are intolerant sites.

## Limitations and conclusions

In this study we investigated the distribution of patient and public variants from three sodium channel databases (Na$_V$1.1, Na$_V$1.2, and Na$_V$1.6) with the intent to link patterns of variation with known physiological functions of these channels. One challenge was the unequal sample sizes in the three patient databases. While the Na$_V$1.1 patient mutations made up roughly 43% of the total Na$_V$1.1 mutations analyzed and was adequate for our analyses, the patient mutations only accounted for 28% and 24% total mutations in Na$_V$1.2 and Na$_V$1.6 databases respectively. While often giving similar OR, these smaller samples sizes did not allow us to assess statistical significance for some segments (i.e. S4 and S6).

We also point out that the Na$_V$1.2 and Na$_V$1.6 databases included cases of varying severity (e.g., from benign cases to those with developmental and epileptic encephalopathy), For the Na$_V$1.1 database, we were able to separate missense variants Dravet Syndrome patients from patients with milder forms of epilepsy (n = 83). While the trends are similar across segments,

linkers, and loops; the enrichment of mild variants in termini contributed to a marginally significant difference in the cumulative distribution as measured by the Anderson-Darling test (p = 0.055) (S3 Fig). Larger databases are needed to distinguish between mild and severe cases of Na$_V$1.2 and Na$_V$1.6 and to distinguish differences in patient variant burden in these channels, notably at the termini. An important implication is that this type of analysis can help to infer the pathogenicity of variants of unknown significance.

In summary, we have found associations that confirm and extend understanding of the function of these three channels. We highlight channel-specific sensitivities within the inactivation gate and S5-S6 linker as well as neuron-specific sensitivities that directly relate to the gain or loss of function within these channels. This study highlights the importance of genotype-phenotype associations at the level of channel function and points to the need to perform more informative studies analyzing epilepsy severity in patients with these channelopathies.

## Supporting information

**S1 Fig. Cumulative distribution plot comparing public and patient Na$_V$1.2 variants.** Brown lines indicate protein boundaries.
(DOCX)

**S2 Fig. Cumulative distribution plot comparing public and patient Na$_V$1.6 variants.** Brown lines indicate protein boundaries.
(DOCX)

**S3 Fig. Cumulative distribution plot comparing mild and Dravet Syndrome Na$_V$1.1 variants.** Brown lines indicate protein boundaries.
(DOCX)

**S4 Fig. Intolerance graph.** The–log p-value of the computed z-scores for the nine VGSCs was calculated and were grouped based on their values. The sodium channel number represents the SCN gene to which each value corresponds.
(DOCX)

**S1 Table. List of all patient and public (GnomAD) mutation numbers for each Na$_V$ channel.**
(DOCX)

**S2 Table. List of patient variants used for these analyses.**
(DOCX)

**S3 Table. Breakdown of regions and their functionality based on previous literature and the ultimate grouping they were placed in.**
(DOCX)

**S4 Table. Statistically significant (p<0.05) OR values for patient variant burden by domain for Na$_V$1.1, Na$_V$1.2, and Na$_V$1.6.**
(DOCX)

**S5 Table. Na$_V$1.1 mutations by segment and domain for the A. patient database and for the B. public database.**
(DOCX)

**S6 Table. Na$_V$1.2 mutations by segment and domain for the A. patient database and for the B. public database.**
(DOCX)

**S7 Table. Na$_V$1.6 mutations by segment and domain for the A. patient database and for the B. public database.**
(DOCX)

## Acknowledgments

We thank Wishes for Elliott and The Cute Syndrome Foundation for family advocacy on behalf of the SCN8A Registry. The authors also acknowledge the Genome Aggregation Database (gnomAD) and the groups that provided exome and genome variant data to this resource. A full list of contributing groups can be found at https://gnomAD.broadinstitute.org/about.

## Author Contributions

**Conceptualization:** Michael F. Hammer.

**Data curation:** Alejandra C. Encinas, Michael F. Hammer.

**Formal analysis:** Alejandra C. Encinas, Joseph C. Watkins, Iris Arenas Longoria.

**Funding acquisition:** Joseph C. Watkins.

**Methodology:** Joseph C. Watkins, Michael F. Hammer.

**Project administration:** Michael F. Hammer.

**Supervision:** Joseph C. Watkins, Michael F. Hammer.

**Writing – original draft:** Alejandra C. Encinas, Michael F. Hammer.

**Writing – review & editing:** Alejandra C. Encinas, Joseph C. Watkins, J. P. Johnson, Jr, Michael F. Hammer.

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
