## [Decision Letter · Decision Letter 0]

19 Jun 2020

PONE-D-20-07729

Variable patterns of mutation density among NaV1.1, NaV1.2 and NaV1.6 point to channel-specific functional differences associated with childhood epilepsy

PLOS ONE

Dear Dr. Hammer,

Thank you for submitting your manuscript to PLOS ONE. After careful consideration, we feel that it has merit but does not fully meet PLOS ONE’s publication criteria as it currently stands. Therefore, we invite you to submit a revised version of the manuscript that addresses the points raised during the review process.

The manuscript needs an expanded discussion along the lines of reviewer 1's notes, with particular emphasis on how this potentially translates into clinical practice and particularly the emerging field of precision medicine.

We look forward to receiving your revised manuscript.

Kind regards,

Randall Lee Rasmusson

Academic Editor

PLOS ONE

Journal Requirements:

'We thank Xenon Pharmaceutical, Inc for providing financial support for a graduate research assistantship to ACE..'

'National Institutes of Health grant GM084905 for a graduate research fellowship to ACE (https://www.nih.gov/)

National Science Foundation grant 1740858 to JCW (https://www.nsf.gov/)

The funders played no role in the study design, data collection and analysis, decision to publish, or preparation of the manuscript'

Additionally, because some of your funding information pertains to [commercial funding//patents], we ask you to provide an updated Competing Interests statement, declaring all sources of commercial funding.

In your Competing Interests statement, please confirm that your commercial funding does not alter your adherence to PLOS ONE Editorial policies and criteria by including the following statement: "This does not alter our adherence to PLOS ONE policies on sharing data and materials.” as detailed online in our guide for authors  http://journals.plos.org/plosone/s/competing-interests.  If this statement is not true and your adherence to PLOS policies on sharing data and materials is altered, please explain how.

Please include the updated Competing Interests Statement and Funding Statement in your cover letter. We will change the online submission form on your behalf.

'The authors have declared that no competing interests exist.'  

We note that one or more of the authors are employed by a commercial company: Xenon Pharmaceuticals

5.Please provide additional details regarding participant consent.

In the ethics statement in the Methods and online submission information, please ensure that you have specified (a) whether consent was informed and (b) what type you obtained (for instance, written or verbal, and if verbal, how it was documented and witnessed).

If your study included minors, state whether you obtained consent from parents or guardians.

If the need for consent was waived by the ethics committee, please include this information.

Reviewers' comments:

Reviewer's Responses to Questions

**Comments to the Author**

1. Is the manuscript technically sound, and do the data support the conclusions?

Reviewer #1: Yes

Reviewer #2: Yes

2. Has the statistical analysis been performed appropriately and rigorously? 

Reviewer #1: Yes

Reviewer #2: Yes

3. Have the authors made all data underlying the findings in their manuscript fully available?

Reviewer #1: Yes

Reviewer #2: Yes

4. Is the manuscript presented in an intelligible fashion and written in standard English?

Reviewer #1: Yes

Reviewer #2: Yes

5. Review Comments to the Author

Reviewer #1: This paper compared patterns of variation in patients’ and public databases to test the hypothesis that regions of known functional significance within the main voltage-gated sodium channels show an increased burden of deleterious variants. The authors demonstrated that patient variant density was higher in regions known to play a role in channel function, e.g. the pore region, which is quite intuitive. On the other hand, they did also found that channel-specific patterns of patient burden may reflect different roles played by the NaV 1.6 inactivation gate in action potential propagation, and by NaV 1.1 S5-S6 linkers in loss of function and haploinsufficiency.

Main comments

This is an original study that could potentially impact the interpretation of the variants in the genes involved in different epilepsy types, in particular, to infer the pathogenicity of variants of unknown significance. The manuscript is well written and it is technically a quite sound piece of scientific research with data that partially supports the conclusions. Experiments have been conducted rigorously, with appropriate sample sizes. The statistical analysis has been performed rigorously.

This reviewer has the following comments:

1) The authors assessed the mutational burden in different regions of the NaV channels considering (1) the excess variants in the public databases and (2) the cumulative distribution of variants throughout the different genes. However, this approach is quite arbitrary. Moreover, there was a clear discrepancy in the choice of the different patients’ database used for this study (i.e., Japanese patients with Dravet syndrome (DS); the literature data for the NaV1.2 missense variants, the patient’s registry for SCN8A mutations. This choice must be discussed as it did potentially affect the results, at least in part. For instance, Nav 1.1 mutations are associated with a spectrum of severity, ranging from autosomal dominant GEFS+ to de novo mutations causing DS. Thus, as honestly discussed by the authors, the adoption of the Japanese register did exclude a priori all the milder form of epilepsy associated with mutations in SCN1A.

2) Overall, pathogenic variants in these genes have been associated with a wide spectrum of epilepsy phenotypes, ranging from benign epilepsies to epileptic encephalopathies with variable severity. Furthermore, a few patients with intellectual disability (ID) or movement disorders without epilepsy have been reported (see, for instance, Johannesen KM, The spectrum of intermediate SCN8A-related epilepsy. Epilepsia. 2019;60(5):830-844).

3) The case of NaV 1.1 variants is even more complicated if we consider that some variants can promote inclusion of a "poison" exon that leads to reduced amounts of full-length SCN1A protein, a mechanism is likely to be broadly relevant to human disease (Carvill GL, Aberrant Inclusion of a Poison Exon Causes Dravet Syndrome and Related SCN1A-Associated Genetic Epilepsies. Am J Hum Genet. 2018;103(6):1022-1029).

Reviewer #2: There are no overall issues with the data or conclusions. The authors clearly describe the data bases used in the statistical analysis and point out the potential shortcomings. It is an interesting approach to identifying differences between the various sodium channels with respect to the relationships between the function/role of the sodium channel in neuronal firing and the region/effects of a given mutation. This approach would allow researchers to study more clinically-relevant mutations in ion channels and give greater insight into the relationship between ion channel structure/function and consequences on neuronal activity.

6. PLOS authors have the option to publish the peer review history of their article (what does this mean?). If published, this will include your full peer review and any attached files.

Reviewer #1: No

Reviewer #2: No

---

## [Author Response · Author response to Decision Letter 0]

2 Aug 2020

Reviewer #1: 

1a) The authors assessed the mutational burden in different regions of the NaV channels considering (1) the excess variants in the public databases and (2) the cumulative distribution of variants throughout the different genes. However, this approach is quite arbitrary. 

Response: We not sure what the reviewer means by “arbitrary”; however, the cumulative function distribution contains all the information necessary for the statistical tests (i.e., sufficient for all the statistical tests performed in these analyses). 

1b) Moreover, there was a clear discrepancy in the choice of the different patients’ database used for this study (i.e., Japanese patients with Dravet syndrome (DS); the literature data for the NaV1.2 missense variants, the patient’s registry for SCN8A mutations. This choice must be discussed as it did potentially affect the results, at least in part. For instance, Nav 1.1 mutations are associated with a spectrum of severity, ranging from autosomal dominant GEFS+ to de novo mutations causing DS. Thus, as honestly discussed by the authors, the adoption of the Japanese register did exclude a priori all the milder form of epilepsy associated with mutations in SCN1A.

Response: We completely agree and thank the reviewer for noting the discrepancy. We have now performed an analysis of ALL SCN1A variants—including an additional 94 patients with milder symptoms in addition to the the Dravet Syndrome patients. While including the milder patients did not alter our main results in a significant way (i.e., variants in the milder group tended to be found in similar segments, domains, linkers and loops as variants with Dravet Syndrome), we did discover that variants associated with these patients tended to accumulate more in the N- and C- terminus. This raises new questions about the functionality of these regions of the gene, which we now discuss in the last section of the manuscript.

Future analyses that include greater numbers of variants will have additional statistical power to identify sites in the channel with patient burden. These studies will need to include associations with patient phenotype (i.e., along the spectrum of disease outcome). 

2) Overall, pathogenic variants in these genes have been associated with a wide spectrum of epilepsy phenotypes, ranging from benign epilepsies to epileptic encephalopathies with variable severity. Furthermore, a few patients with intellectual disability (ID) or movement disorders without epilepsy have been reported (see, for instance, Johannesen KM, The spectrum of intermediate SCN8A-related epilepsy. Epilepsia. 2019;60(5):830-844).

Response: We now refer to this broader spectrum and cite the Johannessen et al. reference. And restate the last sentence of the first paragraph of the introduction to clarify this point.

3) The case of NaV 1.1 variants is even more complicated if we consider that some variants can promote inclusion of a "poison" exon that leads to reduced amounts of full-length SCN1A protein, a mechanism is likely to be broadly relevant to human disease (Carvill GL, Aberrant Inclusion of a Poison Exon Causes Dravet Syndrome and Related SCN1A-Associated Genetic Epilepsies. Am J Hum Genet. 2018;103(6):1022-1029).

Response: We agree that this is an interesting finding. However, we feel it is beyond the scope of this study our focus on variants across the entire gene in patient and public databases while variants in the abovementioned exon occur outside the annotated coding regions.

Reviewer #2: 

There are no overall issues with the data or conclusions. The authors clearly describe the data bases used in the statistical analysis and point out the potential shortcomings. It is an interesting approach to identifying differences between the various sodium channels with respect to the relationships between the function/role of the sodium channel in neuronal firing and the region/effects of a given mutation. This approach would allow researchers to study more clinically-relevant mutations in ion channels and give greater insight into the relationship between ion channel structure/function and consequences on neuronal activity.

---

## [Decision Letter · Decision Letter 1]

11 Aug 2020

Variable patterns of mutation density among NaV1.1, NaV1.2 and NaV1.6 point to channel-specific functional differences associated with childhood epilepsy

PONE-D-20-07729R1

Dear Dr. Hammer,

We’re pleased to inform you that your manuscript has been judged scientifically suitable for publication and will be formally accepted for publication once it meets all outstanding technical requirements.

Kind regards,

Randall Lee Rasmusson

Academic Editor

PLOS ONE

Additional Editor Comments (optional):

Reviewers' comments:

Reviewer's Responses to Questions

**Comments to the Author**

1. If the authors have adequately addressed your comments raised in a previous round of review and you feel that this manuscript is now acceptable for publication, you may indicate that here to bypass the “Comments to the Author” section, enter your conflict of interest statement in the “Confidential to Editor” section, and submit your "Accept" recommendation.

Reviewer #1: All comments have been addressed

2. Is the manuscript technically sound, and do the data support the conclusions?

Reviewer #1: Yes

3. Has the statistical analysis been performed appropriately and rigorously? 

Reviewer #1: Yes

4. Have the authors made all data underlying the findings in their manuscript fully available?

Reviewer #1: Yes

5. Is the manuscript presented in an intelligible fashion and written in standard English?

Reviewer #1: Yes

6. Review Comments to the Author

Reviewer #1: The paper has been improved by implementing the discussion. It is a nice addition to the literature and can support genotype-phenotype correlations.

7. PLOS authors have the option to publish the peer review history of their article (what does this mean?). If published, this will include your full peer review and any attached files.

Reviewer #1: **Yes: **Pasquale Striano

---

## [Editor Report · Acceptance letter]

12 Aug 2020

PONE-D-20-07729R1 

Variable patterns of mutation density among Na_V_1.1, Na_V_1.2 and Na_V_1.6 point to channel-specific functional differences associated with childhood epilepsy 

Dear Dr. Hammer:

I'm pleased to inform you that your manuscript has been deemed suitable for publication in PLOS ONE. Congratulations! Your manuscript is now with our production department. 

Kind regards, 

on behalf of

Dr. Randall Lee Rasmusson 

Academic Editor

PLOS ONE